# A Drift-Aware Clustering and Recovery Strategy for Surface-Deployed Wireless Sensor Networks in Ocean Environments

**DOI:** 10.3390/s25185883

**Published:** 2025-09-19

**Authors:** Lei Wang, Qian-Xun Hong

**Affiliations:** Department of Electrical Engineering, Feng-Chia University, Taichung City 40724, Taiwan; leiwang@fcu.edu.tw

**Keywords:** wireless sensor networks, ocean monitoring, cluster based, dynamic topology, drift-aware routing, disconnection recovery

## Abstract

Wireless sensor networks (WSNs) are deployed in terrestrial environments. However, on the sea surface, sensor nodes can drift due to ocean currents and wind; thus, network topologies continuously evolve, and the communication between nodes is frequently disrupted. These unstable connections significantly degrade data transmission stability and overall network performance. These problems are particularly significant in maritime regions where the sea state changes rapidly, thus imposing stringent technical requirements on the design of long-range, reliable, low-latency, and persistent sensing systems. This study proposes a wireless sensor network architecture for sea surface drifting nodes, which is termed Drift-Aware Routing and Clustering with Recovery (DARCR). The proposed system consists of three major components: (1) an enhanced dynamic drift model that more accurately predicts node movement for realistic ocean conditions; (2) a cluster-based framework that prevents disconnection and minimizes delay, which improves cluster stability and adaptability to dynamic environments through refined clustering and route setup mechanisms; and (3) a self-recovery routing strategy for re-establishing communication after disconnection. The proposed method is evaluated using ocean current data from the Copernicus Ocean Data Center simulating a 60-h drifting scenario around the central Taiwan Strait. The experimental results show that the average hourly disconnection rate is maintained at 6.2%, with a variance of 0.31%, and the transmission of newly sensed data is completed within 3 to 5 s, with a maximum delay of approximately 10 s. These findings demonstrate the feasibility of maintaining communication stability and low-latency data transmission for sea surface WSNs that operate in highly dynamic marine conditions.

## 1. Introduction

As international tensions continue to escalate, nations worldwide are actively investing in the development of automated surveillance systems to allow quick-deployment, long-duration, low-human-intervention intelligence gathering and tactical awareness missions in high-risk areas. Beyond military defense applications, countries in the Pan-Pacific region use behavioral tracking to monitor the distribution of marine fish populations. The real-time monitoring of fish movement patterns and distribution significantly aids fisheries management and ecological research. The long-term observation of specific species’ migration routes can assist in establishing fishing moratoriums and in designating conservation areas.

Marine disasters, such as tsunamis, storm surges, and underwater earthquakes, pose substantial threats to coastal regions; thus, high-frequency, low-latency monitoring systems are indispensable. Effective disaster warning mechanisms reduce personnel and property losses and act as a critical foundation for strengthening national security. Wireless sensor networks are highly scalable, low-cost, and have flexible deployment characteristics and are well suited to these scenarios.

Wireless sensor networks are more flexible and practical for marine surveillance applications such as satellite observation or vessel deployment than traditional monitoring methods. Satellites cover a broad area of observation, but they are constrained by orbital trajectories and weather conditions; thus, the continuous monitoring and the detection of subtle changes and small targets are challenging. Vessel patrols and manual observation can provide more detailed in situ data, but they often involve labor-intensive requirements, high costs, and operational risks; therefore, it is impractical to maintain large-scale deployment and real-time reporting over extended periods.

The advantages of wireless sensor networks are low cost, ease of deployment, and high scalability; therefore, the flexible configuration of sensor nodes is possible, in line with mission requirements, thus establishing continuous monitoring architectures in specific marine areas. These systems collect various marine environmental information for extended periods and can be integrated with other monitoring systems to provide real-time, comprehensive, and accurate information. These systems offer highly practical and deployment-flexible solutions for tsunamis and storm surges; fish distribution and migration route tracking; and security applications, including three-dimensional marine surveillance and defense reconnaissance [1,2].

Traditional wireless sensor networks deploy nodes in fixed location over time. It is assumed that, once deployed, nodes remain stationary; hence, all design objectives focus on reducing network energy consumption to maximize network lifetime and on minimizing data transmission latency for the required coverage. However, if wireless sensor networks are deployed in dynamic marine environments, topology changes caused by environmental factors lead to frequent communication disconnections between nodes [3,4]. Therefore, network lifetime is no longer the only primary design consideration; the focus shifts to constructing wireless sensor network architectures that are resistant to topology changes and incorporate disconnection recovery mechanisms.

Current marine sensor network designs often use anchored deployment to prevent dynamic drift in sensor nodes due to ocean currents [5,6]. This approach limits the movement range or may prevent environmental drift. While this strategy effectively reduces drift-induced challenges, it imposes significant limitations on applicability. In marine environments with depths of several thousand meters, anchored sensor deployment not only introduces severe challenges for recovery and maintenance but also hinders observations of shallow ocean layers and the atmospheric domain above. For military and defense applications, anchored sensors also increase the likelihood of detection and destruction; thus, anchored deployment is impractical for our study target.

This study confronts the challenges of dynamic environments. Regarding the overall system’s cost requirements, the communication modules on nodes cannot support direct data transmission back to a base station; therefore, a cluster-based design [7] is used, whereby sensor nodes are divided into multiple clusters and one node acts as the cluster head to aggregate data from its member nodes before transmitting these data to the base station via multi-hop routing. A cluster-based design is also used because it is well established in traditional wireless sensor network research; it simply requires appropriate adjustments for dynamic marine scenarios and the addition of a disconnection recovery mechanism.

Regarding cost, deployment feasibility, and application scope, the proposed design uses sensor nodes that passively drift with ocean currents on the sea surface and which have no self-propulsion capability and are not constrained by a fixed deployment boundary. The proposed system adopts a cyclic recovery and redeployment mechanism that determines deployment and retrieval locations based on regional ocean current characteristics. When deploying in a marine area with currents flowing from south to north, nodes are deployed in the southern region, recovered at specific northern positions after drifting, and then recharged and redeployed in the south. This approach significantly reduces deployment complexity and decreases cost through cyclical reuse. This design also adapts to seasonal changes in ocean currents, so deployment efficiency and data coverage are greater. These nodes also feature simple structures and low energy consumption; thus, mass production and rapid deployment and scalability are possible.

Even small-scale damage to the system can be quickly mitigated due to the nodes’ deployment and drifting characteristics with ocean currents. In defense applications, they are also resilient to localized enemy destruction. Naturally, floating sensor nodes may be maliciously retrieved or accidentally drift outside the monitoring area; however, anchored nodes are fixed targets when their positions are exposed, and floating nodes are positionally uncertain; thus, precise targeting and destruction are difficult. They can also be camouflaged as floating debris to lower the likelihood of detection.

The system’s cyclical recovery and redeployment mechanism and the simplified node design and massive deployment strategy also mitigate the effect of individual node losses in the overall system. Floating deployment is not entirely risk-free, but it provides a more balanced solution regarding flexibility, deployment cost, and survivability. Cyclical deployment is highly practical and cost-effective, but the relative positional changes in sensor nodes during drift cause communication path interruptions and data loss and remain a major challenge. Data transmission delays that are caused by establishing multi-hop networks must also be considered.

To conquer the challenges mentioned above, this study increases inter-node stability and reduces disconnections to less than a specific threshold through massive deployment using cluster-based multi-hop network topology, minimizes hop counts to limit data transmission time within specified standards, and promptly restores disconnected routes. This study uses an improved drift prediction model and develops a clustering and route setup mechanism that prevents disconnections and reduces latency using drift prediction results. Nodes with smaller predicted movement are selected as cluster heads, and routes are established using hop count and stability. Relay nodes are also used to determine routes for disconnected nodes and resolve disconnection issues.

The remainder of this paper is organized as follows: Section 2 reviews related work, Section 3 details the system model and proposed methods, Section 4 presents simulation settings and an experimental analysis, and Section 5 provides conclusions and suggestions for future work.

## 2. Related Work

Traditional wireless sensor networks focus on maintaining specific coverage rates, extending network lifetime, and minimizing data transmission delays [8]. Few studies concern inter-node connection stability in dynamic environments. Traditional WSNs cannot be directly applied in dynamic environments, but they are a significant reference.

Most wireless sensor networks use a cluster-based design [9], whereby sensor nodes are divided into multiple clusters, and in each cluster, one node serves as the cluster head (CH) and receives, aggregates, and forwards data that are collected by intra-cluster nodes to the base station or upper-layer nodes. A cluster-based architecture reduces the redundant communication between nodes and the overall energy consumption and increases the network lifetime and data transmission efficiency. CHs are selected using competitive and non-competitive approaches. Non-competitive selection directly assigns specific nodes as cluster heads using the base station [10]. Non-competitive selection uses a centralized architecture; thus, it is less suitable for large-scale networks. Competitive selection allows nodes to elect cluster heads based on their individual conditions, such as remaining energy, frequency of serving as cluster head, and distance to a base station [11,12]. This distributed network architecture does not require base station assistance and is suited to large-scale networks where not all nodes can communicate directly with the base station.

N. Xu et al. proposed a method that uses a backoff mechanism and suppression signals for cluster head selection [13], which balances the load across nodes. This study team proposed a distributed architecture for large-scale networks that is called Keeping Desired QoS by a Partial Coverage Algorithm for Cluster-Based Wireless Sensor Networks (KDPC) [14,15]. Experiments confirm its ability to maintain certain coverage rates in static environments while balancing node loads and extending network lifetime. For cluster formation, KDPC employs a competitive mechanism, whereby each node calculates its backoff time based on the previous load and executed rounds and then begins countdown. Nodes with higher energy have shorter backoff times.

Nodes that complete countdown first, without receiving cluster head announcement packets, become cluster heads and broadcast announcement packets. Nodes that receive announcement packets become member nodes and join appropriate cluster heads. A node may receive multiple cluster head announcement packets and selects the cluster head with the strongest signal strength. If a node completes countdown but surrounding cluster heads are full, it enters sleep mode because it is unnecessary in the current round. This mechanism reduces cluster head concentration and resource waste, increases the spatial uniformity of clusters, and reduces data transmission costs between nodes and cluster heads.

The base station initiates the gradual construction of a multi-hop network to establish a routing. It first broadcasts, and cluster heads that receive the broadcast become first-layer cluster heads and establish connections with the base station. These first-layer cluster heads then broadcast HOP packets to search for cluster heads without established routes. HOP packets contain current hop count information. A cluster head may receive multiple HOP packets from cluster heads at different layers, but it establishes connections with the cluster head that has the minimum hop count. When connections are established, cluster heads increase the HOP packet hop count by one and broadcast HOP packets outward. This process continues until all cluster head connections are established and a multi-hop routing with reasonable hop counts is created.

In highly dynamic marine environments, nodes experience positional drift due to factors such as ocean currents and wind; hence, there are frequent topology changes. This study uses drift prediction models to increase network stability. Two primary models are used to describe object movement on the sea surface: the leeway model and the dynamic drift model. The leeway model decomposes wind-induced motion effects on objects into two components: perpendicular and parallel to the wind direction. Different windward surfaces cause leeway to be either left or right of the wind direction. Coefficients in the equation are derived by the linear regression of experimentally obtained data.

Different objects have different floatation characteristics on the sea surface (volume above sea level, windward surface area); thus, dedicated field experiments for specific objects and substantial data are required to regress accurate motion models [16]. The dynamic drift model, which is derived from drag equations, assumes that forces from ocean currents that act on the submerged portion of objects and wind forces that act on the surface portion maintain a static force equilibrium but does not consider the effect of wind on different windward surfaces. It is a universal prediction method that does not require extensive regression analysis; therefore, it is particularly suitable for new sensor nodes or special objects when sufficient historical drift records are not available [17].

Jinfen Zhang et al. used the dynamic drift model and proposed a prediction model [18] that uses a spatial correlation of flow fields to determine object velocity that is derived from a weighted average of current and wind speeds around drifting objects and which gives accurate results. However, the model only uses the dynamic drift model for simulation; thus, it has limitations for actual marine environments.

For WSNs that are deployed on water surfaces, R. Katsuma et al. assumed that nodes are equipped with solar panels and motors to allow autonomous movement; hence, these nodes do not drift from sensing positions due to water flow [19]. However, this method is expensive and does not consider sensor node failures or losses; therefore, targeted destruction renders it inoperable. This study assumes that nodes that do not feature autonomous mobility drift unpredictably.

In underwater wireless sensor networks (UWSNs), the UWSN nodes typically feature one of three configurations: cable-anchored deployment, fixed to the seabed, or autonomously mobile [20,21,22]. Regarding node cost and complexity, most studies focus on only coastal nearshore areas and are unsuitable for large-scale marine monitoring. The scope, the extent of topology changes, and the inter-node communication methods also differ from those used in this study. Nodes that are deployed underwater experience smaller current velocities than those that are deployed on the surface; hence, topological challenges are smaller [23].

This study uses sensor nodes that are deployed on the sea surface that do not move autonomously and have an unrestricted drift range; thus, the topology varies considerably and real-time transmission requirements are problematic. These scenarios are rare in existing WSN and UWSN research. This study proposes a novel framework with disconnection-preventive clustering, routing, and recovery mechanisms that are well suited to a dynamic sea surface sensor network.

## 3. System Model and Proposed Method

The proposed method is called Drift-Aware Routing and Clustering with Recovery (DARCR). Its architecture combines an improved dynamic drift prediction model, stability-oriented cluster-based mechanisms, and routing recovery strategies if there are communication disconnections. The environmental assumptions and node capability settings for this study are as follows:(1)The marine environment simulation for this study only considers the forces of ocean currents and wind and ignores wave effects because object dimensions are much smaller than wavelengths [24].(2)Time resolution is 1 h, assuming that flow field velocities remain unchanged within each hour.(3)Regarding cost and complexity, sensor nodes do not feature autonomous mobility.(4)Sensor nodes are equipped with GPS to determine their coordinate positions.(5)Sensor nodes have two different communication radii: the communication radius R_c1 between member nodes and cluster heads and the communication radius R_c2 between cluster heads and a sensing radius, R_s.

These simplifications are made to balance model clarity and computational feasibility. However, they may introduce deviations under certain sea conditions. In particular, rapid sea-state transitions, gust fronts, or short-term wave-induced disturbances can lead to local variations in flow conditions within the 1 h interval. Additionally, fine-scale dynamics, such as eddies or frontal zones, may introduce spatial and temporal variability that is not captured by the constant-flow assumption. Using shorter time steps (e.g., 10–15 min) and higher-resolution ocean data could help resolve these transient behaviors. Moreover, a shorter time step may affect simulation outcomes, as it allows the model to more accurately capture short-term variations in flow fields and node trajectories. Therefore, we adopt a 1 h time step as a practical trade-off, while considering finer time-resolution models as a promising direction for future work.

This study involves periodic network reorganization, whereby sensor nodes return detected data only when targets are detected. The concept enables node deployment at one end of a marine area and recovery at the other end for cyclical reuse; thus, cluster heads count the current total number of members and report their own positions to the base station before periodic reorganization. The approximate distribution of nodes can then be determined.

### 3.1. Prediction Model

The dynamic drift model provides a relatively simplified treatment of wind effects and uses wind speed and ocean current velocity to determine object movement direction. In reality, the windward surface of objects on the sea surface is not always the same. The angle between the windward surface and the object determines the direction in which the object moves. The leeway model allows the different windward surfaces of objects to produce movement in different directions. Physical sensors and various parameters for the leeway model cannot be measured by extensive field experiments, and there is a time cost for data collection; therefore, this study improves the dynamic drift model by incorporating the advantages of the leeway model. This subsection derives the dynamic drift model and explains the leeway model and how it is integrates and modifies the prediction model.

#### 3.1.1. Original Dynamic Drift Model

Using the derivation process of the dynamic drift model by Röhrs et al., the drag equation [25] is expressed as follows:(1)$$F_{D}=\frac{1}{2}\rho v^{2}C_{D}A$$
where ρ represents the fluid density, v denotes the fluid velocity relative to the object, $C_{D}$ represents the drag coefficient as a dimensionless quantity, and A denotes the reference area. If an object drifts on the sea surface, the portion above sea level experiences wind forces and the portion below sea level experiences ocean current forces. Therefore, under static equilibrium conditions, Equation (2) is written as follows:(2)$$\frac{1}{2}\rho_{w}A_{w}C_{w}\left|U_{w}-U_{o}\right|\left(U_{w}-U_{o}\right)=\frac{1}{2}\rho_{c}A_{c}C_{c}\left|U_{c}-U_{o}\right|\left(U_{c}-U_{o}\right)$$
where subscript w represents wind, subscript c represents ocean current, $U_{w}$ represents the wind velocity, $U_{c}$ represents the ocean current velocity, and $U_{o}$ represents the object velocity. Assuming $\sqrt{\rho_{w}A_{w}C_{w}/\rho_{c}A_{c}C_{c}}=\alpha ,$ after rearrangement, Equation (3) is expressed as follows:(3)$$U_{o}=\frac{1}{1+\alpha }U_{c}+\frac{\alpha }{1+\alpha }U_{w}$$

Setting f=1/(1+α), this equation can be rewritten as Equation (4):(4)$$U_{o}=fU_{c}+(1-f)U_{w}$$

#### 3.1.2. Extension of the Dynamic Drift Model with Leeway Considerations

To explain the leeway model, the assumptions of Kui Zhu et al. state that if wave effects are neglected [16], the leeway model can be written as follows:(5)$$U_{o}=U_{F}+L$$
where $U_{F}$ represents the flow-induced speed, which is typically approximated as the surface ocean current velocity, and L (leeway) represents the drift of objects due to wind and wave effects. Based on previous assumptions, L only represents wind-induced effects. The effect of wind on objects is related to the windward surface; thus, the movement direction due to wind speed on objects is not necessarily parallel to the wind direction (note that this aspect is not considered by the dynamic drift model). L is decomposed into two components: the downwind leeway component (DWL) parallel to the wind direction and the crosswind leeway component (CWL) perpendicular to the wind direction. The relationships between these vectors are shown in Figure 1.

To derive the expression for the angle  θ, L is determined. $U_{F}$ can be approximated as the ocean current velocity $U_{c}$; thus, the following expression for L is obtained from Equations (4) and (5):(6)$$L=f(U_{w}-U_{c})$$

The DWL component is then obtained by projecting the leeway vector onto the wind vector $U_{w}$ via the inner product:(7)$$L_{d}=\frac{\langle L,U_{w}\rangle }{{\left|U_{w}\right|}^{2}}U_{w}$$

The CWL is obtained as follows:(8)$$L_{c}=L-L_{d}$$

Therefore, from Equations (7) and (8), the angle θ is expressed as follows:(9)$$\theta =sign(L\times U_{w})\tan^{-1}\left(\frac{\left|L_{c}\right|}{\left|L_{d}\right|}\right)$$

If θ is calculated, the wind velocity component in the dynamic drift model is corrected using a rotation matrix. Therefore, Equation (4) can be rewritten as follows:(10)$$U_{o}=fU_{c}+\left(1-f\right)U_{w}R(\theta )$$

#### 3.1.3. Improved Prediction Model

Unless there are special circumstances, neither ocean currents nor wind fields exhibit significant changes in velocity and direction between closely spaced locations. Under stable field conditions, current and wind velocities exhibit slow spatial variation with smoothness and correlation. Therefore, regarding this spatial continuity characteristic, the weighted average of ocean current or wind velocity information from neighboring areas is used to improve the stability and accuracy of drift prediction, and prediction deviations due to single-point velocity anomalies are avoided. Özgökmen et al. used a Gaussian function to parameterize the degree of correlation between any two locations [26], which is expressed as follows:(11)$$B(X^{i},X^{j})=exp\left\{-\frac{\left|X_{i}-X_{j}\right|}{2R^{2}}\right\}$$
where $X^{i}$ and $X^{j}$ represent two different locations, and R is a scale parameter that is used to measure spatial correlation. A smaller R value indicates that the degree of correlation between two locations decreases rapidly with distance, and a larger R value indicates a slower decrease. The object velocity prediction model of Jinfen Zhang et al. is described in reference [18]. Let $X_{t-1}={\left[X_{t-1}^{1},X_{t-1}^{\prime }\right]}^{T}$ be a vector, and $X_{t-1}^{1}\;$ is a random variable that features a normal distribution. $X_{t-1}^{\prime }={\left[X_{t-1}^{2},X_{t-1}^{3},\ldots X_{t-1}^{p}\right]}^{T}$ represents grid point locations with correlation degrees of more than 0.3 and is calculated using Equation (11) in the vicinity of the mean object position $x_{t-1}^{1}$ at time t-1, totaling p-1 points. The object velocity is also a random variable that features a normal distribution; thus, ocean current velocity and wind velocity can be obtained by optimal estimation as follows:(12)$$E\left[U_{t}^{1}|U_{t}^{\prime }=u_{2}\right]{=u}_{t}^{1}+\sum_{t}^{12}{\left(\sum_{t}^{22}\right)}^{-1}(u_{2}-u)$$
where $\sum_{t}^{ij}=B(X_{t}^{i},X_{t}^{j})\sigma_{t}^{i}\sigma_{t}^{j}$ denotes the covariance matrix, and $\sigma_{t}^{i}$ denotes the standard deviation of $X_{t}^{i}$, i = 1, 2, 3,…, p. When the current and wind velocities are calculated, the object velocity is calculated using the original dynamic drift model Equation (4). However, in Equation (10), the $R\left(\theta \right)\;$ term is missing; thus, let R(θ) also be a random variable with a normal distribution, which is calculated using Equations (6)–(9). The predicted object velocity and position for the next time step is expressed as follows:(13)$$E\left[U_{o}\right]=\;fE\left[U_{c}\right]+\left(1-f\right)E\left[U_{w}R(\theta )\right]$$(14)$$E\left[X_{t}^{1}\right]=E\left[X_{t-1}^{1}\right]+E\left[U_{t-1}^{o}\right]∆t$$
where ∆t represents the prediction time step if the velocity remains constant within the time step. Although this assumption does not fully capture real-world dynamics, object velocity is estimated using spatial correlations by referencing nearby regional current and wind velocities; therefore, the assumption of constant velocity within the time step is valid.

### 3.2. Challenges and Improvements for KDPC

KDPC divides WSN construction into two phases: cluster establishment and routing establishment. To achieve a specific coverage rate in each round, the number of non-cluster head nodes (NCHNs) that is required for each cluster is calculated as follows:(15)$$N=\frac{log(1-C_{0})}{log(1-\frac{\pi {R_{s}}^{2}}{\pi {R_{c1}}^{2}})}$$
where $C_{0}$ denotes the required coverage rate.

Algorithm 1 is designed for cluster establishment and Algorithm 2 for routing establishment.
**Algorithm 1:** Cluster assembly.1.BS broadcasts a Round Beginning packet with the expected coverage ratio to all nodes2.For all nodes in the wireless sensor network3.Calculate its backoff time for competing4.Backoff time count down5.While (Backoff time is not expired)6.If (there is a CH declaration packet received)7.Save the CH info into the Priority Queue using the signal strength8.End while9.If (Backoff time expired and there is no CH declaration packet received)10.Node announces itself as a CH by broadcasting a CH declaration packet with radius R_c2_11.Calculate the amount of NCHNs, say *N*, needed for the coverage ratio12.While (*N* ≠ 0)13.If (there is a JOIN message sent from a Node, say Node_i_, received)14.Assign a time slot for the new NCHN15.Reply with a PERMISSION message with the slot number to the node Node_i_16.*N* − 117.End if18.Broadcast a COMPLETION message to declare the cluster is assembled19.Else20.While (Priority Queue is not empty)21.Dequeue to get the info of CH22.Send a JOIN message to the CH23.While (Waiting time is not expired)24.If (PERMISSION message is received)25.Receive and record the time slot assigned26.End if27.End while28.End while29.Sleep in this round30.End if31.End Algorithm

**Algorithm 2**: Route setup.1. BS broadcasts a level 1 HOP packet with the radius R_c2_2. While (Waiting time T_a_ is not expired)3.  For all CHs in the wireless sensor network 4.   If (receive HOP packet)5.    If (the HOP level is smaller than current minimum)6.     Save the sender’s information and HOP level7.    End if8.   End if9.   If (waiting time T_b_ is expired and a level *n* HOP packet is recorded)10.    Confirm upstream connection to the saved sender and become level *n*11.    Broadcast a new HOP packet with level *n* + 1 to neighbors12.   End if13. End while14. End Algorithm

KDPC establishes a WSN that maintains specific coverage rates and balances node loads to extend network lifetime. Cluster establishment and route setup reduce and evenly distribute energy consumption. However, in dynamic marine environments wherein real-time transmission is necessary, adjustments must be made. The associated challenges and solutions are described below:


**Challenge** **1: Selected CHs are unstable; thus, disconnection occurs.**


KDPC uses backoff time to select CHs, but the backoff time formula only considers historical load; therefore, the selected CHs can frequently disconnect if movement is excessive.


**Improvement:** The backoff time formula is adjusted to incorporate the predicted node velocity as follows:




(16)
$$\begin{array}{cc} backoff\;time= & T_{1}\times \frac{R_{execute}}{R_{current}}\\ & +T_{2}\times \frac{\sum_{k=1}^{R_{current}}\left(\frac{N_{pass_{through\left(k-1\right)}}}{N_{cluster\left(k-1\right)+}N_{pass_{through\left(k-1\right)}}}\right)}{R_{execute}+1}+T_{3}\times \left|U_{o}\right|\\ & +T_{4}\times Rand[0,1] \end{array}$$



$R_{execute}$: The number of hours a node has been active (serving as a CH or member node).

$R_{current}$: The total number of hours executed.

$N_{pass\_through(k)}$: The total number of packets passing through the node in the k-th hour if the node is a CH; otherwise, the number is 0.

$\left|U_{o}\right|$: The magnitude of the predicted drift velocity.

$T_{1},T_{2},T_{3},T_{4}$: Weights.

Nodes with higher velocities (i.e., less stable) are assigned longer backoff times, reducing the likelihood that they are selected as CHs.



**Challenge 2**
**: Member nodes are prone to drifting from their associated cluster.**



When member nodes decide which cluster to join, they select the CH from the queue with the strongest signal strength. Other CHs may not have the highest signal strength, but their distance to the node may be decreasing and the distance to the CH with the strongest signal strength may be increasing, which will result in disconnection.


**Improvement:** The priority is modified to consider the relationship between signal strength and relative movement:




(17)
$$\;priority=P_{1}\times \frac{RSSI}{{RSSI}_{Rc1}}+P_{2}\times \frac{\left(dist-{dist}_{predict}\right)}{dist}$$



RSSI: The received signal strength for the announcement packet.

${RSSI}_{Rc1}$: The signal strength if the distance is d.

dist: The distance between the node and the CH that sends the announcement packet.

${dist}_{predict}$: The predicted distance after 1 h between the node and the CH that sends the announcement packet.

$P_{1},P_{2}$: Weights.



**Challenge 3**
**: Established routes are prone to disconnection.**



The routing establishment method allows CHs that directly receive BS broadcasts to become Level 1. However, if the CHs that receive broadcasts are already on the verge of disconnection, the newly established routes fail immediately. As these CHs are Level 1, there are many downstream CHs, and disconnections have a significant impact. The method for selecting upstream connections also relies solely on the minimum hop count to determine whether to establish connections. Reducing the hop count decreases data transmission delays, similarly to the Level 1 establishment. If disconnection occurs immediately after establishment, an additional disconnection recovery overhead is required. Therefore, connection targets cannot be determined solely based on the minimum hop count.


**Improvement:** A link score is introduced to evaluate connections to different CHs as follows:




(18)
$$link\;score=L_{1}\times \frac{1}{HOP\_count}+L_{2}\times \frac{\left(Dist-{Dist}_{predict}\right)}{Dist}$$



HOP_count: The hop count for the received HOP packet.

Dist: The distance between the CH and the CH that sends the HOP packet.

${Dist}_{predict}$: The predicted distance after 1 h between the CH and the CH that sends the HOP packet.

$L_{1},L_{2}$: Weights.

Before sending HOP packets, CHs first calculate their positions after 1 h using predicted velocities and include current and predicted positions in the HOP packets. CHs that receive HOP packets calculate the score for each upstream candidate using the link score and select the CH with the highest score as their upstream CH; thus, there is a trade-off between the hop count and disconnection prevention. However, if ${Dist}_{predict}$ is greater than the inter-cluster communication radius, it is excluded.

### 3.3. Disconnection Recovery

Disconnections are unavoidable; thus, before transmitting any data, a connection confirmation packet must first be sent. If no response is received within a specific time period, this is determined as a disconnection, and disconnection recovery is initiated. The detailed procedure is presented in Algorithm 3.The CH that first detects the disconnection broadcasts to search for other nearby non-disconnected CHs. If a non-disconnected CH is found, it establishes a connection and updates the HOP route. If there are no other non-disconnected CHs, the CH broadcasts using backoff time to select appropriate relay nodes and set TTL = 1.

These relay nodes then broadcast (TTL−1) to search for non-disconnected CHs and return the search results, their coordinates, and their own IDs to the CH. If relay nodes find connections, a link is established, but if no connection is found at this point, the CH selects the relay node that is closest to the BS based on the coordinates of these relay nodes.

The CH broadcasts this node’s ID and sets TTL = 2. Nodes that do not have this ID stop searching. The remaining relay node follows the process that the CH’s relay node uses for selection, and this relay node broadcasts to determine a second relay node, and the second relay node expands the search range. If both stages of searching fail to find connections, the CH waits for a period of time before attempting to reconnect.

Relay nodes extend the search range; therefore, nodes that are relatively close to the edge within the cluster must be selected as relay nodes. However, these relay nodes must not be positioned at the critical point for the CH’s communication range because a node may discover an available CH and then immediately lose connection with the relay node during data transmission. If the distance from the CH to a node exceeds 0.9 times the $R_{c1}$, that node is not allowed to participate in the relay node selection process. The backoff time calculation is calculated as follows:(19)$$backoff\;time=B_{1}\times \frac{RSSI}{{RSSI}_{Rc1}}+B_{2}\times Rand[0,1]$$

RSSI: The received signal strength for the broadcast.

${RSSI}_{Rc1}$: The signal strength if the distance is $R_{c1}$.

$B_{1},B_{2}$: Weights.

To allow traditional WSNs to be applied in highly dynamic marine environments, the proposed method, DARCR, uses a drift prediction model that allows the WSN construction process to use prediction results and minimize disconnections. The drift prediction model combines the dynamic drift model with the leeway model to compensate for the dynamic drift model’s limited consideration of wind-induced effects and modifies the prediction model that was proposed by Jinfen Zhang et al. The challenges for traditional WSNs in dynamic marine environments are used to propose corresponding solutions that modify the KDPC architecture.

The selected CHs are insufficiently stable and prone to disconnection; therefore, the backoff time formula is modified, and CH selection considers the node’s past load and the magnitude of the predicted velocity, thus selecting only stable CHs.

Member nodes also easily drift from their originally assigned clusters, and the original priority queue uses a new priority calculation method that simultaneously considers signal strength and the relative movement between nodes and CHs, and nodes choose which CH to join based on current distance and whether the node will move farther from or closer to the CH over the subsequent time period.

Established routes are prone to disconnection; thus, the determination method for Level 1 CHs must prevent CHs that are already at the edge of the communication range from becoming Level 1, which would cause immediate disconnection after a short time and prevent large amounts of data from passing through this CH and being transmitted. The multi-hop routing establishment method is modified to use the hop count to determine connection targets, and a link score evaluates the connection stability of each CH to minimize disconnections and maintain reasonable hop counts.

Despite these disconnection prevention adjustments during WSN construction, disconnections may still occur; therefore, a disconnection recovery algorithm that uses relay nodes to search for non-disconnected CHs using a two-phase search process is proposed.
**Algorithm 3:** Disconnection recovery.CH broadcasts a Connection Check packet to next-hop nodeIf (ACK received within time limit)Proceed with data transmissionElseDisconnection confirmedCH uses communication range $R_{c2}$ to broadcast “Find-CH” packetIf (connected CH is found)Establish new HOP route to that CHUpdate route tableExitCH broadcasts Relay Request using range $R_{c1}$If (Nodes receiving the request and distance to CH $<0.9R_{c1}$)Node calculate backoff time based on signal strengthBackoff time count downIf (Backoff time expired and there is no Relay Ready packet received)Node announces itself as a relay node by broadcasting a Relay Ready packetCH broadcasts “Find-CH” using $R_{c1}$ and sets TTL = 1Relay node broadcasts “Find-CH” using $R_{c2}$, with TTL−1If (connected CH is found by relay node)Relay node returns its ID and coordinates to CHCH establishes route via this relayUpdate route tableExitCH broadcasts selected relay node ID which position is closest to BS and sets TTL = 2Other relay nodes abort searchSelected relay node starts a second search:broadcasts Relay Request using range $R_{c1}$Calculate backoff time, Compete, select second-hop relayBroadcast “Find-CH” with $R_{c2}$ and TTL−1If (connected CH found in second hop)Establish link via two relaysUpdate route tableExitElseWait for a period of time and restart recovery process

## 4. Simulation Results

To evaluate the performance of the proposed DARCR method in dynamic marine environments, we conducted multiple sets of simulation experiments focusing on delay, progressive network construction strategies, and disconnection scenarios across varying node densities. All simulations were implemented in Java (Version 23.0.1, Oracle corporation, Redwood Shores, CA, USA), using Visual Studio Code (Version 1.104.1, Microsoft corporation, Redmond, WA, USA) as the development environment. Each scenario was repeated at least five times, and the results were averaged to ensure statistical reliability.

Two baseline protocols are included for comparison: (1) Original KDPC, which supports hierarchical clustering and multi-hop routing but was not originally designed for drift environments, is also used to highlight the limitations of static clustering strategies in dynamic marine contexts. (2) LEACH-ME incorporates mobility-aware clustering through a remoteness factor [27]. Since LEACH-ME assumes a single-hop transmission model that is unsuitable for large-scale drifting scenarios, we retain its clustering mechanism but replace the routing phase with the original KDPC multi-hop strategy to enable a fair comparison under dynamic conditions.

The parameters and settings for the simulation are as follows:

The Taiwan Strait is the simulation area, and sensors are deployed in a rectangular region of 10,000 km^2^. In Figure 2, blue dots represent sensor nodes, red dots indicate nodes that act as CHs during that hour, and BSs are positioned at the internal two points of the trisection points along the longer side and are represented by black dots. The size of the dots does not represent the actual communication range or physical size. This study uses the Global Ocean Physics Analysis and Forecast (PRODUCT 001_024, DOI: 10.48670/moi-00016) of the EU Copernicus Marine Service as the data source for current and wind field information [28].

$R_{s}$: 2 km; $R_{c1}$: 5 km; $R_{c2}$: 20 km.

Simulation duration: 60 h.

Total number of nodes: 2000.

∆t: 1 hr.

f: 0.97.

$T_{1}$: 0.1; $T_{2}$: 0.3; $T_{3}$: 0.4; $T_{4}$: 0.2.

$P_{1}$: 0.6; $P_{2}$: 0.4.

$L_{1}$: 0.6; $L_{2}$: 0.4.

$B_{1}$:0.8; $B_{2}$:0.2.

### 4.1. Delay Analysis

This study uses VHF radio for sensor node communication because it allows long-range transmission and has low infrastructure requirements, which is particularly suitable for remote or maritime applications [29]. To evaluate the E2E delay, we adopted a conservative assumption. Based on a 32-byte packet size and a 4800 bps data rate, the most optimistic per-hop delay would be approximately 53.33 ms. However, practical VHF transmission also involves additional overhead such as connection setup, synchronization, and other delays. Therefore, we did not use the theoretical minimum but instead assumed 1 s per hop as a more conservative and realistic estimate. For instance, if the hop count is 5, the transmission time would be about 5 s. Thus, in this study, the hop count is directly regarded as the estimated delay time for data transmission from nodes to the BS.

Figure 3 shows the average hop count variation per hour. During the first 30 h, the average transmission time is 3–4 s, but as nodes drift farther apart due to ocean currents, the hop count increases to an average of five hops after 60 h. If the node recovery and redeployment cycle is 60 h, the average delay for data transmission is less than 5 s, but if sensors are deployed in different maritime areas, the recovery cycle is different.

Since the assumption of 1 s per hop is already highly conservative, we further conducted a sensitivity analysis to examine the robustness of the estimated E2E delay under different per-hop delays. Figure 4 presents the sensitivity results for hop counts of 3–5, with per-hop delays ranging from 0.2 to 1.0 s. The results indicate that even with variations in per-hop delay, the E2E transmission time in typical scenarios remains within the 3–5 s range, supporting the validity of our delay assumption.

The average hop count and the maximum hop count per hour affect the estimation of target object positions. Figure 5 shows the maximum hop count within each hour. Initially, some nodes’ data are only transmitted after approximately 6 s, but the average for all nodes is only 3–4 s. Within the 60 h period, the maximum hop count is maintained at less than 10 hops; therefore, data transmission from the node farthest from the BS requires less than 10 s.

### 4.2. Performance of a Progressive Construction Strategy

In dynamic environments, communication disconnections between nodes are inevitable. This study uses numerous strategies to prevent disconnections; thus, it is necessary to determine the effectiveness of each improvement on network disconnection rates. A disconnection is considered to occur if a node loses connection with its immediate upper-layer CH or when any link in the hop route is broken. If a CH has ID 2 and CH2’s hop route is [BS-CH7-CH4-CH8], where CH2’s upper layer is CH8, and a disconnection occurs between CH7 and CH4, both CH2 and CH8 count as one disconnection.

Figure 6 shows the hourly disconnection ratio if the original KDPC is used in a dynamic marine environment without any disconnection prevention measures. The average disconnection rate is 25.6%; therefore, on average, one-quarter of the network’s data are detected but are not successfully transmitted. A total of 40% of nodes can experience disconnection.

To provide a mobility-aware baseline, Figure 7 shows the performance of LEACH-ME, which incorporates a remoteness factor into cluster head selection. This helps slightly reduce disconnection compared with KDPC, with an average disconnection rate of 20.3%. However, LEACH-ME does not include dynamic link evaluation strategy, which limits its effectiveness in highly dynamic ocean environments where node positions change continuously.

Figure 8 shows the results if the link score is used; thus, the predicted relative movement between each layer of CHs is calculated before establishing connections. The average disconnection rate decreases to 20%, but significant disconnection can occur because inner-layer CHs connect to many outer-layer CHs; if disconnection occurs between Level 1 CHs and the BS or between Level 1 and Level 2 CHs, many disconnections occur. Restrictions are added for Level 1 CHs: if the prediction indicates that disconnection will occur within 1 h, the node is prevented from becoming Level 1. These results confirm the hypothesis of this study, as shown in Figure 9 for hourly disconnection rates. The average disconnection rate decreases to 8%, and the maximum disconnection rate is less than 20%. If stability is then measured using the magnitude of the predicted velocity during CH selection, as shown in Figure 10, the average disconnection rate decreases to 6.2%.

The results for progressive improvement are shown in Figure 11, wherein the blue line represents the hourly disconnection ratio for the original KDPC, the red line shows the results after adding the link score, the purple line represents the addition of Level 1 restrictions, and the green line shows the results if stability is considered for CH selection.

This experiment reveals a clear trade-off between connection stability and routing distance and supports the inclusion of the link scoring mechanism in our design. In addition to disconnection rate analysis, we further examined whether the link scoring mechanism increases routing distance.

To this end, we conducted a comparative experiment by removing the link score term from the formula and measuring the average hop count with and without the mechanism. As shown in Figure 3 and Figure 12, the average hop count without link scoring is 3.5, which is slightly lower than the score of 3.65 observed when the mechanism is enabled. This indicates that the network tends to select shorter paths when stability is not considered. However, these shorter paths come at the cost of reduced stability and increased disconnection rates.

### 4.3. Effect of Different Node Densities on Disconnection Rates

Between 1000 and 2000 nodes were deployed in the same area to determine disconnection rates for different node densities. Figure 13 shows that as the number of nodes increases from 1000 to 1300, the average disconnection rate stabilizes at less than 8%. Deploying 1000 nodes gives a 4% higher disconnection rate than deploying 1000 nodes. Regarding node cost and system complexity, 1300 nodes give good communication stability, and further increases in node density produce a limited improvement in disconnection rates. To maintain communication stability, approximately 1300 nodes give the optimum result regarding deployment efficiency and stability.

To examine the influence of key design parameters on system behavior, we selected some representative weights (e.g., T3, L1) and tested small perturbations (±0.2) around their default values. The results, shown in Table 1, illustrate how moderate adjustments in these parameters affect the disconnection rate and average hop count.

## 5. Conclusions

This study proposes a WSN architecture called DARCR that can be deployed in highly dynamic marine environments using velocity prediction to prevent disconnections. Regarding practical applications and complexity, nodes cannot move autonomously, and the effect of these improvements on network stability and performance is determined by simulation. For velocity prediction, two models and an improved dynamic drift model are used, and the following improvements are made using KDPC: (1) the predicted velocity is used for the CH selection backoff time formula, thus increasing the stability and energy of nodes to preferentially become CHs and balance load and reduce disconnection; (2) the priority formula is modified in the priority queue so that nodes choose which CH to join using the current signal strength and the relative movement between nodes and CHs to reduce the risk of nodes drifting from clusters; (3) routing construction that uses a link score calculation that allows CHs to balance hop count and connection stability, adding Level 1 CH restrictions to prevent the selection of Level 1 CHs that are already on the verge of disconnection and avoiding massive disconnection after a short period; and (4) a disconnection recovery algorithm that allows CHs that detect disconnection from upper layers to broadcast and search for non-disconnected CHs. If this fails, relay nodes are used with TTL for two-phase searching.

Regarding simulation results, the hop count results show that for a period of less than 60 h, the proposed method maintains an average delay of 3–5 s and a maximum delay of around 10 s.

The results of improvement strategies are demonstrated. Simulation results confirm that each mechanism positively contributes to disconnection prevention, reducing the average disconnection rate from an initial 25.6% to 6.2%. The number of deployed nodes is also varied to determine the effect on disconnection rates. The results show that a node count of 1300 represents a balance point between deployment efficiency and stability.

This study addresses node drift and communication disconnection issues for wireless sensor networks in dynamic marine environments by proposing a construction strategy with a predictive capability and a recovery mechanism that is stable in simulation environments. There are several opportunities for further optimization, such as adding an adaptive weight parameter adjustment mechanism to determine whether to focus on balancing node energy or reducing delay and disconnections. In addition, the current model adopts a fixed 1 h time step and ignores wave-induced disturbances, which are reasonable for large-scale drift but may not fully capture short-term or small-scale dynamics. Future extensions may explore the use of higher temporal resolution and incorporate wave effects when finer-grained behaviors are of interest. No field experiments were conducted in this study. Physical sensor nodes would enable the recording of accurate drift trajectory and drift characteristic data, which would produce a more accurate drift prediction model and a more optimal overall construction strategy.

This study makes multiple improvements to traditional WSNs, including the use of predictive drift, priority adjustments, disconnection recovery procedures, and actual flow field data and the production of a stable network in dynamic environments. The future use of actual marine data and adaptive algorithms will increase the scalability and applicability to practical scenarios, including marine monitoring and disaster response.

## Figures and Tables

**Figure 1 sensors-25-05883-f001:**
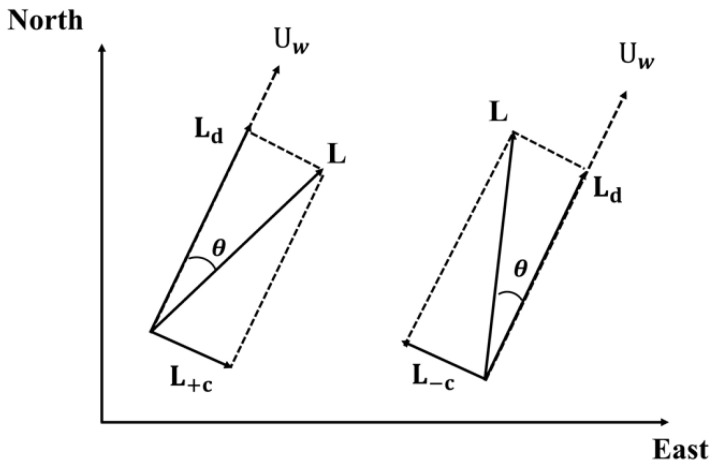
Relationship between leeway vector, wind speed, and DWL and CWL components.

**Figure 2 sensors-25-05883-f002:**
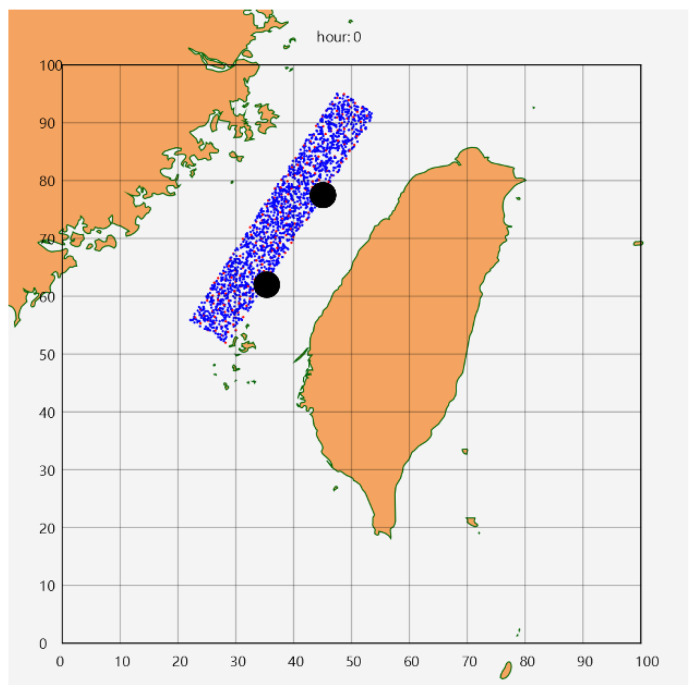
Sensor node deployment.

**Figure 3 sensors-25-05883-f003:**
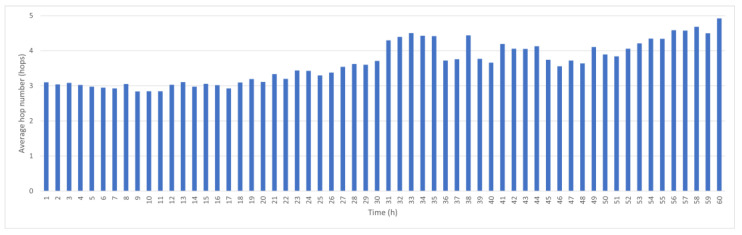
Average hop count per hour.

**Figure 4 sensors-25-05883-f004:**
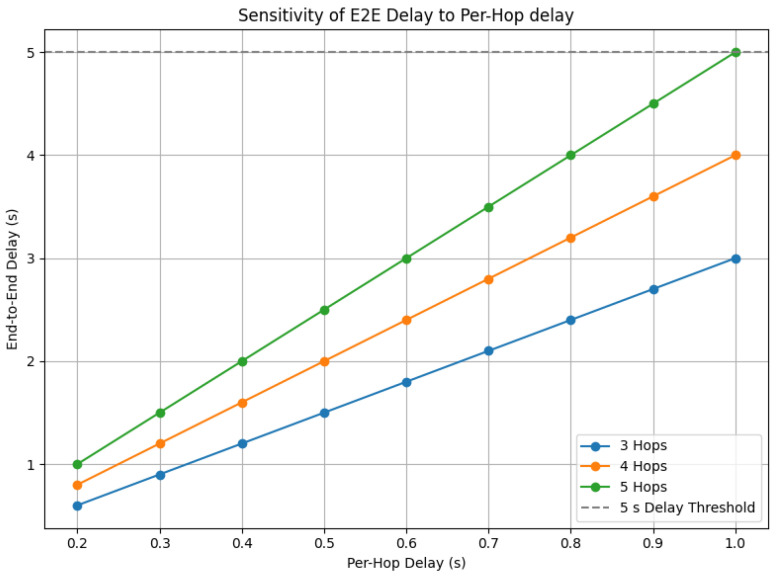
Sensitivity analysis of end-to-end delay with hop counts of 3–5 under varying per-hop delays (0.2–1.0 s).

**Figure 5 sensors-25-05883-f005:**
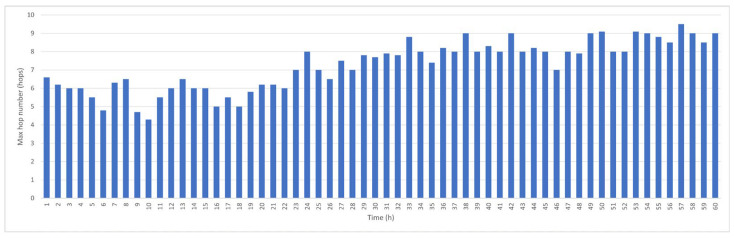
Maximum hop count per hour.

**Figure 6 sensors-25-05883-f006:**
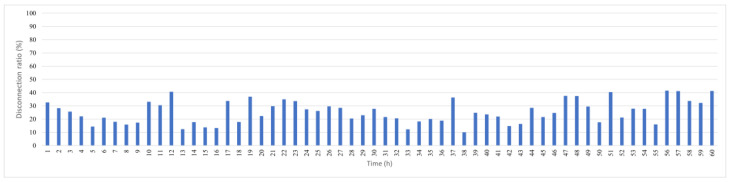
Original KDPC hourly disconnection rate.

**Figure 7 sensors-25-05883-f007:**
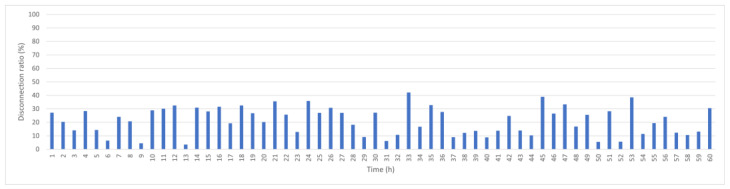
LEACH-ME hourly disconnection rate.

**Figure 8 sensors-25-05883-f008:**
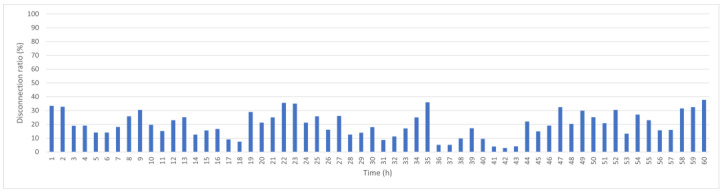
Hourly disconnection ratio after adding link score.

**Figure 9 sensors-25-05883-f009:**
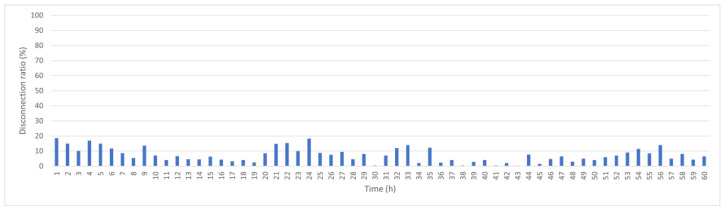
Hourly disconnection ratio after adding Level 1 restrictions.

**Figure 10 sensors-25-05883-f010:**
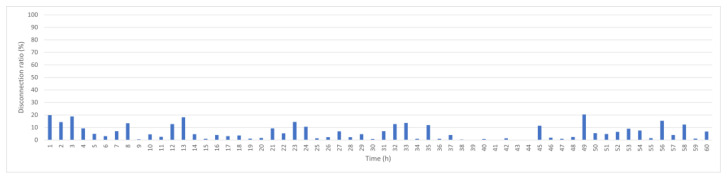
Hourly disconnection ratio if stability is considered for CH selection.

**Figure 11 sensors-25-05883-f011:**
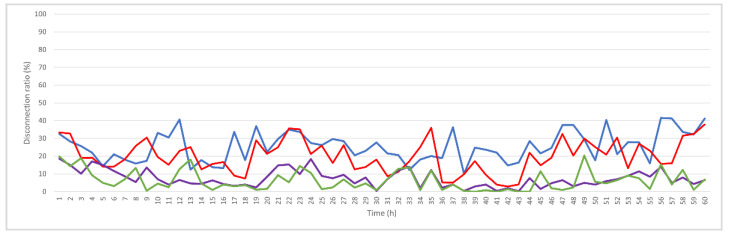
Results for progressive improvements for disconnection ratio without windowing or smoothing.

**Figure 12 sensors-25-05883-f012:**
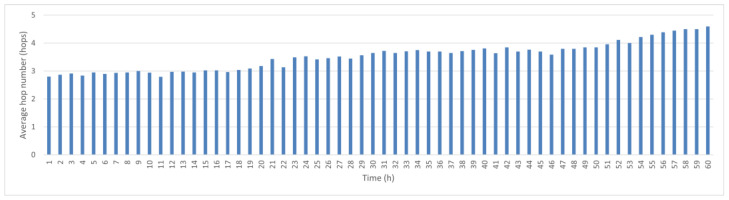
Average hop count per hour without link score.

**Figure 13 sensors-25-05883-f013:**
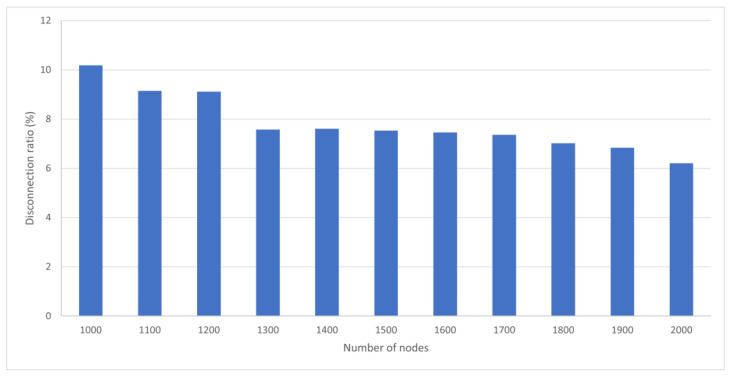
Disconnection rate for different node densities.

**Table 1 sensors-25-05883-t001:** Effects of varying key weight parameters (±0.2) on disconnection and hop count.

Parameter	Weight Setting	Average Disconnection Rate (%)	Average Hop Count (hops)
L1:L2	8:2	6.51%	3.54
L1:L2	**6:4**	**6.20%**	**3.65**
L1:L2	2:8	6.18%	3.80
T3	0.6	6.31%	3.70
T3	**0.4**	**6.20%**	**3.65**
T3	0.2	6.58%	3.63

The original configuration (baseline) is shown in bold.

## Data Availability

This study was conducted using E.U. Copernicus Marine Service Information. All products are available on the Copernicus Marine Service portal at https://marine.copernicus.eu/ (accessed on 3 March 2025). The primary dataset used is the Global Ocean Physics Analysis and Forecast prduct (Product ID: GLOBAL_ANALYSISFORECAST_PHY_001_024; DOI: 10.48670/moi-00016). Data can be accessed via the official Copernicus Marine Data Store, for example: https://data.marine.copernicus.eu/product/GLOBAL_ANALYSISFORECAST_PHY_001_024/description (accessed on 17 September 2025). The simulation uses hourly ocean current data (variables uo, vo) from 1 May 2025 12:00 UTC to 4 May 2025 00:00 UTC.

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
