# Peer review of "A Drift-Aware Clustering and Recovery Strategy for Surface-Deployed Wireless Sensor Networks in Ocean Environments"

_sensors, 2025, doi:10.3390/s25185883_

Round 1

Reviewer 1 Report

Comments and Suggestions for Authors

The manuscript proposes DARCR, a drift-aware clustering and recovery strategy for sea-surface WSNs that integrates:
(i) an improved drift prediction model (dynamic drift + leeway angle), 
(ii) clustering and routing choices that consider predicted motion, and 
(iii) a two-stage disconnection recovery via relay nodes.
Simulations over realistic Copernicus ocean fields in the Taiwan Strait (60-hour window) show low average hop counts (3–5) and markedly reduced disconnection rates versus KDPC-style baselines. The application is timely and the integration of prediction into clustering/routing is a solid idea.

The following points need addressing:

1-In the dynamic drift model,  f=1/(1+α) and must lie in (0,1]. The simulation settings list f=1.05, 
which is physically inconsistent and would invalidate velocity predictions unless this is a typo. Please correct  f (or redefine the model) and re-calculate any dependent results.

2- Eq. (19) for relay-node backoff introduces E and ET  in the legend but they do not appear in the formula. 
Either include the energy term(s) explicitly or remove the unused definitions. Clarify the intended role of energy in relay selection.

3-You assume 1 s per hop for VHF to estimate E2E delay (hop count ≈ seconds). 
Please justify with a brief link-budget/throughput argument or cite measured one-hop delays under the assumed MAC, packet size, and channel conditions. 
Add a sensitivity plot (e.g., 0.2–2.0 s/hop) to show the robustness of the 3–5 s claim.

4- The abstract states an average hourly disconnection rate of 6.6%, whereas the results/conclusions emphasize 6.2% as the final average after progressive improvements. 
Please reconcile and report a single, consistent figure (with variance).

5-Several weights and thresholds (e.g.,  T1-T4, P1, P2, L1, L2, B1, B2, and the 0.9 Rcl  exculsion) appear fixed but their selection process isn’t described. 
Please (i) add a rationale or automated tuning procedure, and 
(ii) include a sensitivity/ablation study to show how results vary with these parameters.

6- KDPC and its progressive modifications are useful, but a stronger case for novelty/benefit requires comparison with standard cluster-based WSN protocols (e.g., LEACH/HEED and mobility-aware variants) configured for your scenario, even if adapted minimally for drift. 
Clarify whether KDPC’s coverage/lifetime goals disadvantage it in dynamic drift and discuss—to avoid a biased baseline.

7- You state each dataset was run “at least 5 times and averaged,” but no dispersion metrics (SD/SE/CI) or significance tests are shown. 
Please report confidence intervals or boxplots for 
(i) hourly disconnection ratio, 
(ii) hop counts, 
and (iii) density sweep (1000–2000 nodes). 
Provide the random seeds/setup and clarify whether ocean fields are fixed or varied per run.

8- The paper uses Copernicus “Global Ocean Physics Analysis and Forecast (PRODUCT 001_024)” but the Data Availability Statement reads “Not applicable.” 
Please provide a citable data source (with version/time span), download instructions, and ideally the scripts to reproduce preprocessing, seeding, and simulation runs.

9- The model assumes 1-hour time steps with constant flows and ignores waves. These are reasonable simplifications but should be highlighted with implications (e.g., sea-state changes, gust fronts). Consider a brief discussion of when these assumptions break and whether shorter timesteps would alter results.

10- The two-phase TTL-based recovery is a good addition, but the added control overhead and its effect on tail latency are not quantified. Please instrument 
(i) control packet counts, 
(ii) extra hops added during recovery, and 
(iii) time-to-recovery distributions.

Some minor issues regarding the language and presentation need to be fixed, for example:
1- A careful English edit is needed (e.g., “JAVA is used for simulation ND Visual Studio Code Is the development environment”, duplicated/awkward sentences in anchored-sensor discussion). 
There are scattered capitalization and spacing issues.
2- Ensure all axes include units (seconds, hops, %, km) and that captions explain the trend (mean vs. max, windowing). Where you report “less than 10 seconds,” reflect that in the y-axis bounds.

Overall, the idea and results are promising; addressing the points above will make the contribution robust and reproducible.

Reviewer 2 Report

Comments and Suggestions for Authors

The paper as a whole has certain academic value, but there is still room for improvement in the argumentation and evaluation of some key details.

Round 2

Reviewer 1 Report

Comments and Suggestions for Authors

The authors have done a great job in addressing the previous comments.
The manuscript should be in great shape once text editing is performed.

Author Response

Response 1: We greatly appreciate the reviewer’s positive comments and kind recognition of our work. To further improve the clarity of the manuscript, we have asked a native speaker to assist with language polishing. All revisions have been marked in red in the revised manuscript to highlight the changes.

Reviewer 2 Report

Comments and Suggestions for Authors

Can Accept

Author Response

Response: We sincerely thank the reviewer for the positive evaluation and constructive feedback. We are encouraged by the reviewer’s recognition of the clarity of our manuscript, and we will carefully keep this standard in the revised version.